# Atovaquone: An Inhibitor of Oxidative Phosphorylation as Studied in Gynecologic Cancers

**DOI:** 10.3390/cancers14092297

**Published:** 2022-05-05

**Authors:** Arvinder Kapur, Pooja Mehta, Aaron D Simmons, Spencer S. Ericksen, Geeta Mehta, Sean P. Palecek, Mildred Felder, Zach Stenerson, Amruta Nayak, Jose Maria Ayuso Dominguez, Manish Patankar, Lisa M. Barroilhet

**Affiliations:** 1Department of Obstetrics and Gynecology, University of Wisconsin-Madison, Madison, WI 53705, USA; a_kaur@yahoo.com (A.K.); mfelder@wisc.edu (M.F.); zstenerson@wisc.edu (Z.S.); 2Department of Materials Science and Engineering, University of Michigan, Ann Arbor, MI 48109, USA; mpooja@umich.edu (P.M.); mehtagee@umich.edu (G.M.); 3Department of Chemical and Biological Engineering, University of Wisconsin-Madison, Madison, WI 53706, USA; adsimmons2@wisc.edu (A.D.S.); sppalecek@wisc.edu (S.P.P.); 4Drug Development Core, Carbone Cancer Center, University of Wisconsin-Madison, Madison, WI 53705, USA; ssericksen@wisc.edu; 5Department of Biomedical Engineering, Macromolecular Sciences and Engineering, Precision Health, Rogel Cancer Center, University of Michigan, Ann Arbor, MI 48109, USA; 6Department of Molecular Genetics and Cell Biology, University of Chicago, Chicago, IL 60637, USA; amrutapn@uchicago.edu; 7Department of Dermatology, University of Wisconsin-Madison, Madison, WI 53715, USA; ayusodomingu@wisc.edu

**Keywords:** metabolism, oxidative phosphorylation, mitochondria

## Abstract

**Simple Summary:**

The Warburg effect in cancer cells (high glucose update and lactate release) is an adaptation that is considered a hallmark of most neoplasms. Blocking oxidative phosphorylation is one way to combat carcinogenesis. Atovaquone is a mitochondrial complex III inhibitor. It is currently FDA-approved for the treatment of malaria, and is a well-tolerated, orally available medication. Our laboratory studied the anti-cancer properties of atovaquone in gynecologic cancers. We found that atovaquone slowed ovarian cancer growth in both cell lines and mouse models. Additional anti-cancer effects were seen, such as the reduced proliferation of cancer stem cells and spheroids implanted in mice. Atovaquone inhibited oxygen consumption and ATP production. Metabolic studies showed that atovaquone shifted glycolysis, electronic transport and the citric acid cycle. Our studies provided the mechanistic understanding and preclinical data to support the further investigation of atovaquone’s potential as a cancer therapy for gynecologic cancers.

**Abstract:**

Oxidative phosphorylation is an active metabolic pathway in cancer. Atovaquone is an oral medication that inhibits oxidative phosphorylation and is FDA-approved for the treatment of malaria. We investigated its potential anti-cancer properties by measuring cell proliferation in 2D culture. The clinical formulation of atovaquone, Mepron, was given to mice with ovarian cancers to monitor its effects on tumor and ascites. Patient-derived cancer stem-like cells and spheroids implanted in NSG mice were treated with atovaquone. Atovaquone inhibited the proliferation of cancer cells and ovarian cancer growth in vitro and in vivo. The effect of atovaquone on oxygen radicals was determined using flow and imaging cytometry. The oxygen consumption rate (OCR) in adherent cells was measured using a Seahorse XFe96 Extracellular Flux Analyzer. Oxygen consumption and ATP production were inhibited by atovaquone. Imaging cytometry indicated that the majority of the oxygen radical flux triggered by atovaquone occurred in the mitochondria. Atovaquone decreased the viability of patient-derived cancer stem-like cells and spheroids implanted in NSG mice. NMR metabolomics showed shifts in glycolysis, citric acid cycle, electron transport chain, phosphotransfer, and metabolism following atovaquone treatment. Our studies provide the mechanistic understanding and preclinical data to support the further investigation of atovaquone’s potential as a gynecologic cancer therapeutic.

## 1. Introduction

Aerobic glycolysis is an important adaptation found in cancer which allows cells to rely on glycolysis to provide the biomolecules needed for biomass production [1,2]. Emerging data demonstrate a complex relationship between glycolysis and oxidative phosphorylation (OXPHOS) in cancer cells [3,4] with mitochondrial oxidative metabolism actively contributing to tumor growth. The four mitochondrial complexes (Complex I–IV) transfer electrons from NADH and FADH_2_ to molecular oxygen. During electron transport, a proton gradient develops in the intermembrane space that drives ATP synthesis. Drugs that inhibit these mitochondrial complexes interfere with metabolism and ATP production and trigger a sharp rise in oxygen radicals, resulting in oxidative damage and cell death [5,6,7]. Developing OXPHOS inhibitors has become an area of interest for cancer treatment and prevention [8,9,10,11]. Our group has demonstrated that products containing unsaturated carbonyl functional groups are inhibitors of OXPHOS. The naphthoquinone plumbagin interferes with mitochondrial electron transport and induces an increase in intracellular oxygen radicals [12]. We identified atovaquone for its close structural similarity to plumbagin and sought to characterize its anti-tumor properties and methods of action in gynecologic cancers.

Atovaquone, an FDA-approved anti-malarial drug, inhibits mitochondrial electron transport. Atovaquone interferes with the ubiquinone-mediated transfer of electrons at mitochondrial complex III [13] and inhibits the proliferation of cancer cells [14]. Atovaquone has shown anti-tumor activity in gynecologic cancer cell lines, but no data on its activity on cancer stem cells, spheroids, or mouse models of gynecologic cancers have been published. We demonstrate the anti-cancer activity of atovaquone in ovarian and endometrial cancers and confirm its mechanism of action through comprehensive molecular and cellular biological experiments, including novel experiments with cancer stem cells, spheroids, and markers of oxidative stress.

## 2. Materials and Methods

### 2.1. Reagents and Cell Lines

Protein detection reagents for Western blotting were from BioRad (Hercules, CA, USA). Buffers and cell media were from ThermoFisher (Waltham, MA, USA). Antibodies were from Cell Signaling Technology (Danvers, MA, USA) and Jackson ImmunoResearch Laboratories (West Grove, PA, USA), respectively. An FITC-Annexin V Apoptosis Detection kit was purchased from BD Pharmingen (San Diego, CA, USA). Reagents for Seahorse and Oroboros experiments were from Agilent Technologies (Santa Clara, CA, USA) and Sigma (St. Louis, MO, USA). ID8 and MOSEC cells were from Dr. Katherine Roby, University of Kansas. Other ovarian cancer cell lines were from ATCC (Manassas, VA, USA) and ECC-1 cells were a gift from Dr. Elaine Alarid (U. Wisconsin–Madison). FITC BrdU cell cycle kit was purchased from BD Pharmingen (559619, San Jose, CA, USA). Atovaquone was purchased from Sigma (St. Louis, MO, USA), purity 99%.

### 2.2. Cell Line Authentication

All previously established cell lines were maintained in the recommended culture media and maintained at 37 °C in a humidified atmosphere with 5% CO_2_. Growth and morphology were monitored and cells were passaged when they reached 80% confluence. Experiments were conducted within one year of the purchase from ATCC or were validated by Single Tandem Repeat (STR) analysis (Genetica DNA Laboratories, Burlington, NC, USA) within three years of conducting the experiments. Mycoplasma testing was performed each month on all cell lines. All experiments were performed on cells free of mycoplasma.

### 2.3. Cell Viability Assays

Cell proliferation in 2D cultures was monitored by the MTT assay as described previously [15]. For 3D culture assays, 2.5 × 10^3^ cells of OVCAR-3, SKOV3 and ECC1 were plated in each well of 96 well ultra-low attachment plates to develop cell spheroids. After 24 h, the spheroids were either treated with DMSO (vehicle) or atovaquone for 72 h. The spheroids were stained with calcein AM and propidium iodide and images were obtained using Zeiss confocal microscope.

Cancer stem-like cells (CSCs) were isolated from two high-grade serous ovarian carcinomas using previously outlined protocols [16,17,18]. Single-cell suspensions were incubated with ALDEFLUOR (ALDEFLUOR assay kit and ALDEFLUOR DEAB, Stem Cell Technologies, Vancouver, BC, Canada), and CD133 antibody (CD133-APC, 130-113-184, and APC-isotype IgG2b, 130-092-217, Miltenyi Biotec, Auburn, CA). ALDH^+^/CD133^+^ cells were isolated using fluorescent activated cell sorting on the Beckman Coulter MoFlo Astrios. Patient-derived CSCs were seeded at a density of 100 cells/well at a density of 100 cells/well on hanging drops plates and monitored for spheroid formation. At day 5, atovaquone was added to the 3D spheroids at 50 µM final concentration. After 72 h, spheroids were imaged and an MTT assay was performed to measure the viability. Flow cytometry (Cytoflex, Beckman Coulter, Indianapolis, IN) enabled the cell cycle analysis using BrdU-FITC and 7-AAD, and was performed on single-cell suspension from the harvested spheroids using manufacturer’s protocol.

### 2.4. Monitoring Apoptosis, DNA Breaks and Oxygen Radicals

Cells were treated with atovaquone or exposed to vehicle (DMSO) for 18 h. Cells were labeled with Annexin V-FITC and propidium iodide or permeabilized and labeled with FITC-conjugated anti-cleaved caspase 3 antibody. DNA damage was monitored in cells treated with atovaquone for 72 h. The cells were permeabilized and labeled with FITC-conjugated anti-phospho-γ-H2Ax antibody, and then monitored by flow cytometry using an Attune (ThermoFisher) cytometer. Data were analyzed using the FlowJo (Ashland, OR, USA) analysis package. The effect of atovaquone on intracellular oxygen radicals was determined using flow and imaging cytometry. Cells were pre-labeled with 10 μM H_2_-DCFDA for 30 min at 37 °C, followed by treatment with atovaquone for 15 min. The cells were washed with PBS, harvested, and intracellular oxygen radical levels were monitored by flow cytometry.

### 2.5. Western Blotting

Cells (3 × 10^5^) were treated with atovaquone or DMSO, washed with ice-cold PBS and lysed in RIPA buffer containing protease inhibitors. Protein concentration was measured using the BCA protein assay (ThermoFisher). Lysates equivalent to 25–30 μg of total protein were electrophoresed, blotted to PVDF membranes and probed with primary and secondary antibodies. The protein bands were detected using chemiluminescence substrate.

### 2.6. Imaging Cytometry

Harvested cells (1 million/mL) were stained with Ghost 780, MitoTracker Green, MitoSox Red and DraQ5 to stain for live/dead determination, mitochondria, mitochondrial oxygen radicals and nucleus, respectively. Cells were treated with atovaquone or vehicle for 15 min, 30 min, 1 h and 2 h then imaged on ImageStream Mark II (Amnis Corporation, Seattle, WA, USA). Data were acquired using INSPIRE V.200.1.388.0 acquisition software, using Channel 1 for Bright Field and Channel 2 (533/55 filter) for green fluorescence, Channel 4 for Mitosox Red, Channel 5 for DRAQ5 to exclude the debris and Channel 6 for Ghost 780, the amine reactive viability dye. The images were acquired at a magnification of 60× with 488 illumination at 5 mW and 785 nm illumination at 1.72 mW laser power. Results from imaging cytometry were analyzed using the proprietary IDEAS V 6.1.303.0 analysis package.

### 2.7. Docking of Atovaquone with Mitochondrial Complexes

Complex I (unliganded mitochondrial complex I (sheep), PDB accession code 5LNK), Complex II (atpenin-bound mitochondrial complex II (wild boar), PDB accession code 3AEE), Complex III (atovaquone-and ubiquinone-6-bound cytochrome bc1 complex (mouse), PDB accession code 4PD4) from RCSB PDB website were assigned polar hydrogens and partial charges with AutoDockTools v1.5.6 [19]. The Smina package, a fork of AutoDock Vina v1.1.2, was used for docking 3D SD format structures of atovaquone [20] and 1,4-naphthoquinone [21] using the default search parameters and scoring function [19]. The autobox ligand feature was applied to specify docking search spaces around ubiquinone reduction sites (Q-sites), a Q-site at Complex II (atpenin-occupied) and the Q_0_ and Q_i_ sites of Complex III (atovaquone and CoQ occupied, respectively). At each site, the co-crystallized ligand was re-docked to validate the docking procedure by reproducing crystallographic poses (<1.0 Å RMSD). In Complex I, the CoQ site is not indicated by a co-crystallized ligand. Therefore, a putative Q-site, based on extensive site-directed mutagenesis data [22] was used: centered at Cartesian position (78.6, 91.8, 197.0) with box edges of 32.0 Å. Docked complexes were inspected and rendered using PyMOL (the PyMOL Molecular Graphics System, Version 2.4.0 Schrödinger, LLC, New York, NY, USA).

### 2.8. Oxygen Consumption Rate

Oxygen consumption rate (OCR) was measured using a Seahorse XFe96 Extracellular Flux Analyzer. Cancer cells (25,000/well) were seeded overnight in XFe96 cell culture plates at 37 °C. Cells were treated with either DMSO or 25 µM atovaquone for 30 min, washed in pre-warmed XF buffer and kept in non-CO_2_ incubator for temperature and pH equilibration. OCR was measured using a pre-optimized concentration of FCCP. Data were analyzed using Wave software.

### 2.9. Mitochondrial Respiratory Complex Activity

The mitochondrial respiratory complex activity was measured on Oxygraph-2K (Oroboros Instruments, Innsbruck, Austria) using the substrate inhibitor titration protocol. Briefly, 2.5 × 10^6^ cells/mL in MIR05 buffer was added to each chamber and baseline oxygen consumption was measured followed by the addition of 16 µM digitonin to permeabilize the cells. Once the oxygen consumption stabilized, DMSO was added to chamber A (control) and 25 µM atovaquone was added to Chamber B (atovaquone-treated cells). After 5 min, Complex I substrates, malate (2 mM) and glutamate (10 mM) were added to initiate the respiration followed by the addition of 5 mM ADP to achieve maximum respiration. Activated Complex I was then inhibited by adding 1.25 µM Rotenone. Complex II/III respiration was stimulated by adding 10 mM succinate (complex II substrate) followed by inhibition with 2.5 μM Antimycin A (Complex III inhibitor). Lastly, complex IV was activated by the addition of 1 mM TMPD (tetramethyl-p-phenylene-diamine) and 0.8 M ascorbate and inhibited by 500 mM sodium azide. Data analysis was performed using the DatLab software 5.1.

### 2.10. NMR Sample Preparation

Intracellular metabolites were extracted from 70–80% confluent cultures of cancer cells grown in one T75 flask per sample in triplicate after 0.25, 0.5, 1, 6, 12, and 24 h of atovaquone (or vehicle) exposure. Cells were washed twice with ice-cold PBS (pH 7.4), quenched and lifted into 6 mL of methanol and stored at −20 °C. Extraction was carried out with a dual liquid phase procedure as described previously [23]. Quenched cells were vortexed in a 10:10:9 volumetric mixture of chloroform:methanol:water and subsequently centrifuged for 15 min at 2000× *g*, resulting in an upper aqueous phase, a lower organic phase and a solid protein interphase. The aqueous phase was transferred to a fresh conical tube, dried at 30 °C in a vacufuge, and stored at −80 °C until analysis. Two-fold excess methanol was added to the remaining organic and protein phases prior to centrifugation for 20 min at 2000× *g*, resulting in a protein pellet which was dried, re-suspended in 500 µL of 1% SDS in H_2_O, and assayed for total protein content via BCA assay according to the manufacturer’s instructions.

On the day of NMR analysis, dried metabolite samples were re-suspended in 600 µL of NMR buffer (100 mM phosphate buffered D_2_O, pH = 7.0 containing 0.5 mM TMSP as an internal reference standard, and 0.2% *w*/*v* sodium azide) and centrifuged for 10 min at 18,000× *g*. Then, 550 µL of supernatant was transferred into 5 mm NMR tubes (Norell Inc, Morganton, NC, USA) and stored on ice until data acquisition.

### 2.11. NMR Acquisition and Processing

^1^H NMR spectra were acquired at 298 K with a Bruker Avance III 500 MHz console (11.74 T) equipped with a 5 mm cryogenic probe with a NOESYGPPR1D pulse sequence. 256 free induction decays (FIDs) were acquired with a spectral width of 16 ppm and an acquisition time of 2 s. Zero-filled data were pre-processed with Bruker TopSpin^TM^ software (version 3.6) with an exponential window function (line broadening, LB = 0.3), Fourier transformation and automatic phase correction. Chemical shifts were referenced to the TMSP peak (δ = 0.00 ppm). Spectra were imported into ACD/1D NMR Processor (ACD Labs, Toronto, ON, Canada) for manual baseline correction and water peak removal (4.75–5.0 ppm). Subsequently, peaks were annotated with ChenomX NMR Suite Profiler (version 7.7), and quantified via reference to the TMSP peak (0.5 mM).

### 2.12. NMR Metabolomic Analysis

The ChenomX output metabolite concentration table was uploaded to MetaboAnalyst. Samples were normalized by total metabolite concentration (to account for differences in extraction efficiencies) and autoscaled (mean-centered and divided by standard deviation). The built-in MetaboAnalyst modules for time-series/two-factor, statistical, and enrichment analyses were utilized to process these normalized and scaled data.

### 2.13. In Vivo Tumor Models and Therapy

IACUC approval (#PRO0009732) was obtained prior to conducting animal experiments. All mice were housed six per cage per institutional standard in our Small Animal Facility. Temperature, humidity, and day/night cycles in the facility are closely monitored. Humane endpoints were used. Adult female C57BL/6 mice (Envigo, Madison WI) were injected intraperitoneally (i.p.) with 1 × 10^7^ MOSEC in 0.2 mL PBS on day 0. Tumor-bearing mice began treatment by oral gavage on day 8 after tumor implantation. Mepron^®^ (GlaxoSmith Kline, Research Triangle Park, NC, USA) was diluted in sterile 2.5% aqueous benzylalcohol and administered (100 µL at 200 mg atovaquone/kg) via oral gavage 5 days/week for 10 weeks. Control tumor-bearing mice received sterile 2.5% aqueous benzyl alcohol at matching volume and schedule. Mice were monitored for any adverse symptoms, the development of palpable tumor, and indications of ascites accumulation. Mice were euthanized as soon as the control group developed significant ascites (approximately 10 weeks from IP injection). Volumes of ascites and tumor weights were recorded at the time of necropsy and tissue collection.

For experiments with patient-derived CSCs, immunodeficient NSG female mice (6–8 weeks old) were purchased from Taconic Biosciences (Rensselaer, NY, USA). Spheroids from patient-derived CSCs were generated as described earlier [16,17,24] and injections were prepared by carefully harvesting spheroids using a pipette and supporting them within Growth-Factor-Reduced Matrigel (Corning, NY, USA). The injections were administered subcutaneously into the mice. Each injection contained 10 spheroids initiated with 60 cell/well that were grown in 3D for 7 days. Tumor size and body weight was measured twice weekly using calipers. Mice with tumors measuring 100 mm^2^ were treated by oral gavage starting on day 30 after tumor implantation. Atovaquone (Sigma Aldrich, St Louis, MO, USA) was diluted in sterile castor oil administered (100 µL at 200 mg atovaquone/kg) via oral gavage 5 days/week for 15 days. Control mice received sterile castor oil at matching volume and schedule. Mice were monitored for adverse symptoms and the development of a palpable tumor. Tumors were allowed to grow until 1500 mm^2^ was reached for maximum tumor burden. Mice were euthanized either at end-point or when they exhibited low body condition scoring (ULAMGSOP End-Stage Illness Scoring System) according to IACUC approved protocol. Tumors were dissected and routine IHC was performed to investigate proliferation and apoptosis status.

## 3. Results

### 3.1. Atovaquone Reduces Viability and Induces Apoptosis in Gynecologic Cancer Cell Lines

Atovaquone inhibited the viability of human ovarian and endometrial cancer cell lines of human (OVCAR-3, SKOV-3 and ECC-1) (Figure 1A) and murine (ID8) origin (Appendix A) by inducing apoptosis as indicated by an increase in annexin V on the cell surface (Figure 1B), cleaved caspase 3 (Figure 1C) and cleaved caspase 9 and a decrease in Bcl-2 (Appendix A). The average IC_50_ was 10 μM which is commensurate with the human plasma concentration when this drug is used for malaria prophylaxis [24]. Atovaquone also increased cell death in 3D spheroids of ECC-1, SKOV-3 and OVCAR-3 cells (Figure 1D). OVCAR-3 cells from loose spheroids were the most affected by atovaquone, presumably because of the higher access of the drugs to the cells in the spheroid (Figure 1D).

### 3.2. Mepron Decreases Tumor Growth in an Orthotopic Model for Ovarian Cancer

Animal experiments were approved by our Institutional Animal Care and Use Committee (protocol ID M005763). To obtain clinically translatable results, Mepron^®^ (the atovaquone formulation prescribed against malarial infection) was tested in an in vivo ovarian tumor model. There was no randomization nor blinding—all mice received active drug. Daily administration of Mepron^®^ by oral gavage was well tolerated by the mice with peritoneal MOSEC tumors. At 10 weeks post implantation of the MOSEC cells, palpable tumors were observed at the site of injection. From this point until the termination of the experiment, the tumor-bearing animals subjected to Mepron^®^ treatment showed a slower rate of weight gain (data not shown). At the time of terminal sacrifice at 14 weeks, four out of five animals showed decreased tumor weights and none to the minimal accumulation of ascites (0–2 mL) (Figure 1E,F). With the one mouse in the test cohort showing no major difference in the tumor weight, the average difference between the test and control cohorts was not statistically significant (*p* value calculated to be 0.1202 by two-tailed unpaired *t*-test). When this mouse was excluded, the data were statistically significant with a *p* < 0.05. The average difference in ascites accumulation in the test versus control cohorts was significant (inset in Figure 1F).

### 3.3. Atovaquone-Induced Cell Death Occurs through the Generation of Intracellular Oxygen Radical Flux

Atovaquone contains the quinone substructure that induces oxidative stress [12] and was found to rapidly increase intracellular oxygen radicals in ECC-1, SKOV-3, OVCAR-3, ID8 and MOSEC cells (Figure 2A). Pretreatment with the radical scavenger N-acetylcysteine (NAC) attenuated the anti-proliferative and pro-apoptotic effect of atovaquone on the cells (Figure 2B,C and Appendix A), indicating a major mechanism by which atovaquone mediates its anti-cancer effects.

We used imaging cytometry to determine the cellular location of the oxygen radicals in response to atovaquone. Exposure to atovaquone (25 µM) resulted in an 81% and 90% increase in mitoSox fluorescence for the ECC-1 and OVCAR-3 cells, respectively, over matching controls (Figure 2D) demonstrating mitochondria as the primary site for the oxygen radicals.

### 3.4. Atovaquone Activates p53

While exploring the effect of atovaquone-induced oxidative stress, we observed an increase in phospho-γH2Ax staining in the treated cells (Figure 3A). The oxygen radicals induced by atovaquone caused damage to the cellular DNA as determined by an increase in the staining with γ-H2Ax (Figure 3A). We tested whether the DNA damage was associated with activation of p53. Longer-term (24, 48 and 72 h) exposure to atovaquone increased the level of total p53 in ECC-1 and OVCAR-3 cells (Figure 3B). These observations were complemented by an increase in the phosphorylation of p53 on its Serine-15 residue, as observed after short duration exposure of the cells to atovaquone. Furthermore, the p53 inhibitor, pifithrin-α, significantly inhibited apoptosis in atovaquone-treated ECC-1 and OVCAR-3 cells (Figure 3C). SKOV-3 cells do not express p53 and therefore were not used in these experiments. In these cells, atovaquone mediates apoptosis via a p53 independent mechanism [12].

### 3.5. Atovaquone Mimics Ubiquinone When Docked to Q-sites of Mitochondrial Complexes I, II, and III

The mitochondrial complexes I, II and III are major sites for oxygen radical formation [25,26]. These complexes contain sites for the electron carrier, CoQ (ubiquinone). As a naphthoquinone, atovaquone likely mimics CoQ to interfere with mitochondrial electron transport. We used molecular docking to examine atovaquone’s complementarity with the various CoQ binding sites (Q-sites) on Complexes I–III in the electron transport chain. In Complexes II and III, atovaquone docking poses were compared with crystallographic positions of benzoquinone-containing ligands at the Q-sites. Atovaquone was docked at the known Q-sites on mitochondrial Complexes II (site Q) and III (sites Q0 and Qi) and a putative Q-site on Complex I based on data from a previous site-directed mutagenesis study looking at the complex function and inhibitor potencies [22]. Favorable atovaquone docking poses show that its napthoquinone substructure overlaps with co-crystallized benzoquinone or benzoquinone-like ligands observed in Q-sites in some of the crystal structures (atpenin in Complex II and CoQ at site Q0 in complex III). The re-docking of atovaquone reproduced the bound pose of co-crystallized atovaquone at the Qi site of Complex III [27,28]. Smaller benzoquinone-containing ligands were also docked at these sites to compare quinone substructure positions. For example, in the atovaquone-bound Q0-site of complex III, the docked poses of 1,4-naphthoquinone, ubiquinone-1, and plumbagin have benzoquinone/naphthoquinone moieties adopting the same position as the naphthoquinone ring of atovaquone (see atovaquone and napthoquinine in Figure 4A (i). At an alternative Qi site in complex III, the positioning of benzoquinone moieties among docked analogues adopted that of the ubiquinone-bound crystal structure (Figure 4A (ii)). Substantial overlap was found in benzoquinone positioning for docked atovaquone and napthoquinone and the pyridinone substructure of atpenin bound in its Complex II crystal structure at Q-site (Figure 4A (iii)). Finally, we considered a putative Q-site near the interface of the PSST and 49 kDa subunits of Complex I [22]. Using the volume encompassing critical residues for inhibitor potencies as the docking search space, atovaquone docked favorably into a pocket near the Fe–S cluster N2 with high overlap with docked napthoquinone (Figure 4A (iv)). This pocket, adjacent to the inner mitochondrial membrane structure, is suitable for accommodating the benzoquinone moiety of the endogenous CoQ. These docking results support the plausibility of a competitive inhibitory mechanism by which atovaquone interferes at multiple Q-sites in the oxidative phosphorylation chain in addition to the Qi site of Complex III.

### 3.6. Atovaquone Inhibits Oxidative Phosphorylation

Pre-incubation (30 min) of ECC-1, OVCAR-3 and SKOV-3 cells with atovaquone decreased basal respiration compared to the control cells (Figure 4A). The uncoupling of the mitochondrial proton gradient by FCCP did not rescue maximal respiration in atovaquone-treated cells (Figure 4A). We also observed that cells treated with atovaquone showed a significant decrease in ATP production (Figure 4A).

We then noted that supplementation with complex I substrates, malate and glutamate or succinate, a complex II substrate, did not reverse the inhibition of oxygen consumption by atovaquone (Figure 4B,C). However, when the atovaquone-treated cells were supplemented with ascorbate/TMPD, the substrates for complex IV, oxygen consumption was restored to the same level as in the untreated control cells (Figure 4B,C). These results indicate that the effect of atovaquone was upstream of complex IV.

### 3.7. Atovaquone Treatment Alters Energy and Amino Acid Metabolism

In order to investigate the broader metabolic impacts of atovaquone treatment, NMR-based metabolomic profiling was performed to identify metabolites enriched in ECC-1, OVCAR3 and SKOV3 cell lysates after exposure to atovaquone treatment for durations ranging from 15 min to 24 h. Principal component analysis (PCA) demonstrated that the extent of deviation of intracellular metabolite profiles between atovaquone-treated and control groups increased with the duration of atovaquone treatment, with statistically significant differences (*p* < 0.05) between drug-treated and control groups observed at 6, 12, and 24 h (Figure 5A). A low total metabolite yield in the 24 h atovaquone exposure group suggested that significant death had occurred, supported by the marked loss of ATP and aforementioned increase in p53 protein levels observed at 24 h. Partial least squares discriminant analysis (PLS-DA) performed on the combined 6 and 12 h treatment groups, at which times atovaquone affected metabolite profiles but did not result in extensive cell death, demonstrated significant separation between the atovaquone treatment and control groups (Figure 5B).

### 3.8. Global Metabolic Perturbations in Response to Atovaquone Treatment of ECC-1 Cells

Metabolites affected by atovaquone treatment, as determined by the Variable Importance in Projection (VIP) scores for principal component 1 (Figure 5B), can be broadly grouped as amino acids (tyrosine, lysine, glutamate, valine, pyroglutamate, isoleucine, ornithine) or energy production/transfer intermediates (creatine phosphate, succinate creatine, AMP, ADP) (Figure 5C). In total, 30 compounds were found to be significantly different in abundance between atovaquone treatment and the control (ANOVA, FDR < 0.05; Figure 5D). Broader metabolite enrichment analysis, performed in MetaboAnalyst, revealed numerous metabolite sets which were differentially enriched between treatment groups as tabulated in Figure 5D. Taken together, the data indicate key shifts in energy (glycolysis, citric acid cycle, electron transport chain, phosphotransfer), amino acid, nucleotide, and urea metabolism following atovaquone treatment.

### 3.9. Atovaquone Inhibits In Vivo Proliferation of Patient-Derived Ovarian Cancer Stem Cell Initiated Tumors

CSC rely on oxidative phosphorylation for their energy needs [28,29]. Therefore, we tested the effects of atovaquone on the viability of the ovarian cancer stem-like cells in 3D spheroids. Spheroids were generated with ALDH+ and CD133+ cancer stem-like cells from two high-grade serous ovarian cancer patients on the 3D hanging drop platform [15,30,31,32]. The incubation of the spheroids with atovaquone in vitro 3-D cultures resulted in 65–75% cell death in atovaquone-treated patient-derived spheroids, as compared to control (Figure 6A,B). The patient-derived CSCs were predominantly arrested in Sub Go/G1 (Figure 6C,D). Ovarian cancer stem-like cell spheroids were subcutaneously implanted in immunocompromised NSG mice and allowed to develop to a size of 100 mm^3^. The daily administration of atovaquone (200 mg/kg) via oral gavage over a 15-day period resulted in a significant reduction in tumor volume (Figure 6E). The immunohistology of the tumors from atovaquone-treated mice demonstrated a significant decrease in nuclear antigen Ki-67 as compared to tumors from control mice (Figure 6F).

## 4. Discussion

Our studies provide the mechanistic understanding and preclinical data to support the use of atovaquone against gynecologic malignancies, particularly ovarian and endometrial cancers. Atovaquone is a potent inhibitor of oxidative phosphorylation in cancer cells. This inhibition results in intracellular oxygen radical flux that is responsible for triggering apoptotic cell death pathways in the cancer cells. We demonstrate that Mepron, the atovaquone formulation administered against malarial infections, inhibits ovarian tumor growth in a syngeneic mouse model. Furthermore, our studies demonstrate that atovaquone inhibits the growth of tumors formed by implanting patient-derived cancer stem cell (CSCs) spheroids in an immunocompromised xenograft mouse model.

Ovarian cancer is often a fatal disease that can become resistant to cytotoxic chemotherapy. Even PARP inhibitors, which have shown great promise in patients with homologous recombination deficiency [33] have their limitations, with many tumors eventually developing resistance [34]. Chemoresistance is a phenomenon particularly observed in CSCs that develop recurring tumors that are resistant to subsequent therapeutic regimens [35]. The high proliferative rates of cancer cells requires them to engage in aerobic glycolysis, while the relatively quiescent CSCs rely on oxidative phosphorylation for their energy needs [36]. Therefore, oxidative phosphorylation is an attractive therapeutic target against CSCs.

In silico experiments suggested that atovaquone can dock to the ubiquinone binding sites of Complexes I–III. Even when cells are supplemented with substrates for Complexes I and II, atovaquone continues to inhibit mitochondrial respiration. Supplementing with Complex IV substrates reverses the inhibitory effect of atovaquone on oxygen consumption. These experiments suggest that although previous studies indicate that Complex III is the primary target [37], it is possible that atovaquone also suppresses Complexes I and II. The major consequence of electron transport inhibition by atovaquone is the increase in intracellular oxygen radical flux (Figure 2). The neutralization of these radicals reverses the cytotoxic effect of atovaquone. The oxygen radicals cause damage to the DNA, as confirmed by the increase in staining for phospho-γ-H2Ax. Our studies also demonstrate that the DNA damage caused by atovaquone activates the tumor suppressor p53, as demonstrated by the increase in phosphorylation of Serine-15. Inhibition of p53 by pifithrin-α attenuates the cytotoxic response of atovaquone in p53 wild type ECC-1 cells [38]. However, OVCAR-3 cells express mutant p53 [39], suggesting that at least a subset of p53 mutants can trigger cell death in response to oxidative damage.

We recently demonstrated that the oxidative damage caused by atovaquone inhibits Na^+^/K^+^-ATPase [12], leading to membrane depolarization and subsequent cell death [40]. Thus, the expression of a wild-type or functional mutant p53 is not obligatory for atovaquone-mediated cytotoxicity. These observations support the cytotoxicity observed in the p53 negative SKOV-3 cells.

The NMR metabolomics data confirm the inhibition of oxidative phosphorylation by atovaquone. Time course measurements indicated, however, that the cancer cells engage in aerobic glycolysis following exposure to atovaquone. Since oxidative phosphorylation is a major metabolic pathway for CSCs, our experiments with patient-derived cells indicate that atovaquone therapy will be especially effective at targeting these tumor initiating cells. To manage chronic oxidative stress, cancer cells rely on anti-oxidant mechanisms through HIF-1α and Nrf-2, the two transcription factors that control anti-oxidant responses [41]. Compared to normal cells, cancer cells have a limited capacity to neutralize the rapid oxidative flux induced by atovaquone. In ongoing experiments, we are testing the combined use of atovaquone with Nrf-2 inhibitors to circumvent effective anti-oxidative responses. Potential synergy may also be realized by combining atovaquone with PARP inhibitors. PARP inhibitors are expected to prevent the DNA repair mechanisms that are engaged following oxidative stress. There are many opportunities to explore atovaquone’s anti-cancer effects in ovarian, endometrial and other cancers.

## 5. Conclusions

In conclusion, we demonstrated that atovaquone is a strong candidate for the treatment of gynecologic tumors. This agent has an acceptable toxicity profile based on preclinical in vitro and in vivo data. Future studies will focus on developing a cancer-specific formulation of atovaquone and the combination of this drug with inhibitors of anti-oxidant mechanisms.

## Figures and Tables

**Figure 1 cancers-14-02297-f001:**
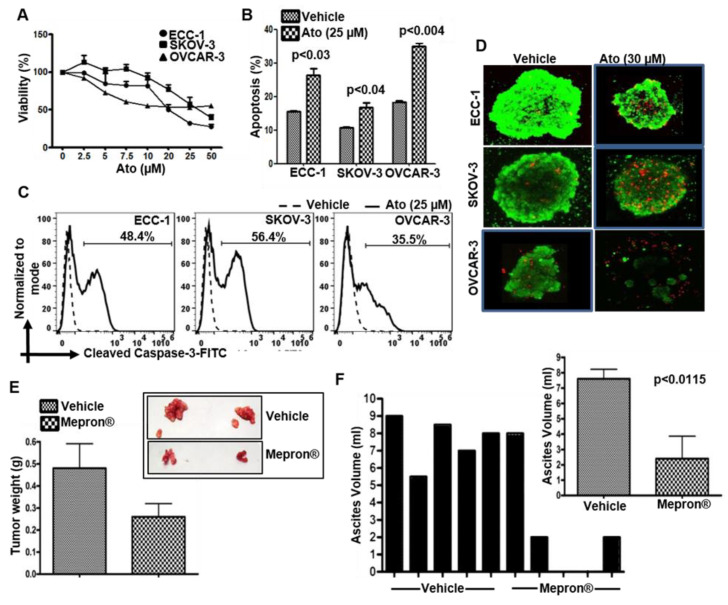
Atovaquone inhibits the proliferation of gynecologic tumors. (**A**) Effect of atovaquone (Ato) on viability of ECC-1, SKOV-3 and OVCAR-3 cells was tested in MTT assays. Average data from three biological repeats with 8 technical repeats in each experiment are shown. (**B**) Cells treated with vehicle (DMSO) or atovaquone were monitored for Annexin V and propidium iodide staining by flow cytometry. The Annexin V single positive and Annexin V and propidium iodide double positive events were recorded as percent apoptosis. Data from the three replicates are shown. (**C**) Apoptosis in the atovaquone-treated cells was confirmed by the monitoring for cleaved caspase-3 by flow cytometry. Plots are representative of three biological repeats. (**D**) Spheroids treated with vehicle (DMSO) or atovaquone were stained with calcein-AM and propidium iodide to monitor viability and cell death. Images are representative of three biological repeats. (**E**,**F**) C57BL/6 mice with intraperitoneal MOSEC tumors were randomly assigned to two groups (N = 5/group). Five tumor-bearing mice were randomly placed in the each of the two cohorts. Mice in the treatment group were dosed with Mepron^®^ five days a week for 10 weeks. Mice in control group were identically dosed with vehicle. When the mice in the vehicle control group were moribund, all of the animals were necropsied and the weights of the excised tumors were measured (**E**) along with the volume of the accumulating ascites fluid (**F**). Inset in **E** shows excised tumors from two mice in the control and atovaquone treatment groups. Inset in **F** shows average volume of ascites recovered from the control and atovaquone treated tumor-bearing mice.

**Figure 2 cancers-14-02297-f002:**
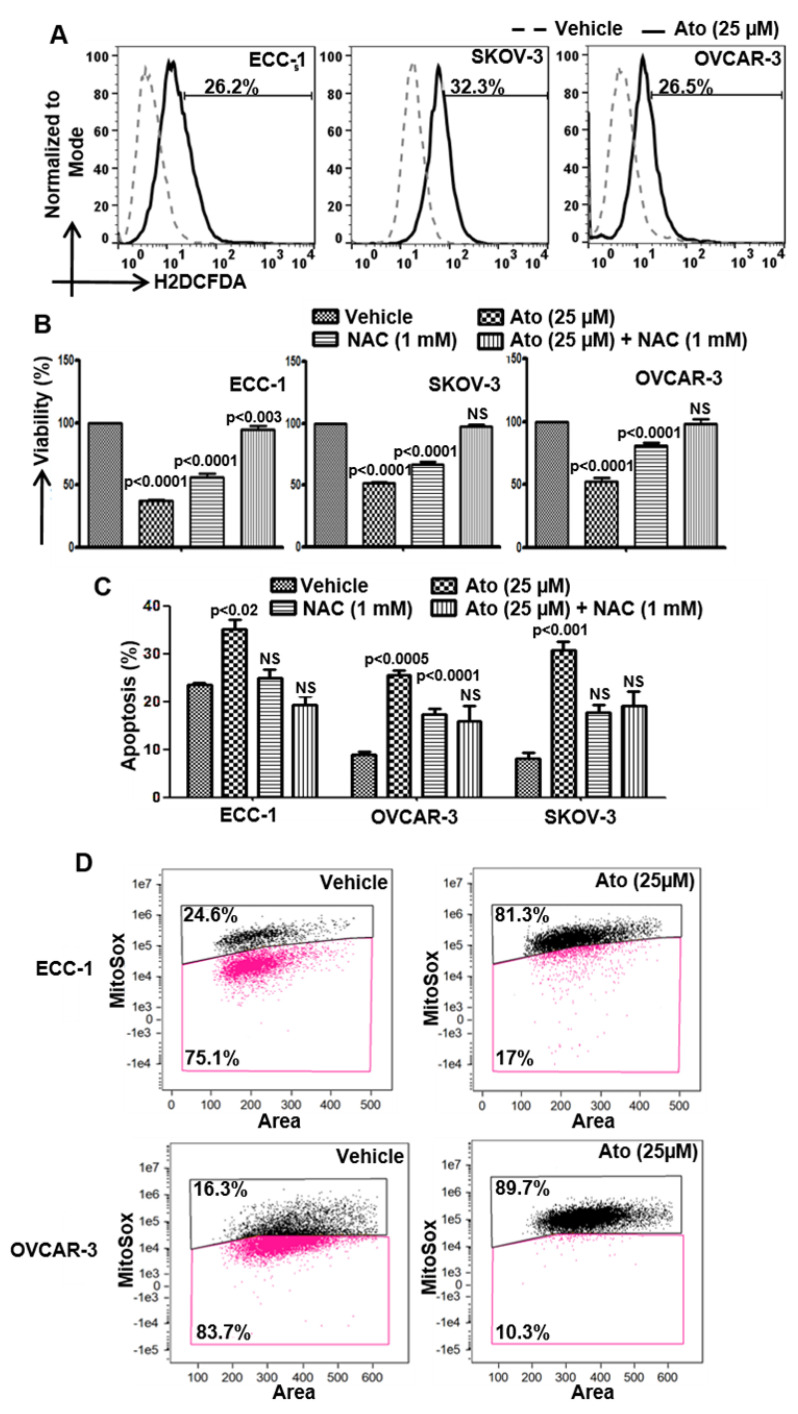
Atovaquone induces oxidative stress in cancer cells. (**A**) Cells labeled with H2DCFDA were exposed to atovaquone (Ato) or DMSO (vehicle) and the intracellular oxygen radical flux was determined using flow cytometry. (**B**,**C**) Cells were preincubated with N-acetylcysteine (NAC) for 30 min prior to treatment with Ato and viability was monitored using the MTT assay and apoptosis by flow cytometry for Annexin V-FITC and propidium iodide. (**D**) Cells were pre-labeled with DRAQ5, Mitotracker green and Mitosox. After treatment with Ato or DMSO (vehicle), the radical flux was monitored by imaging cytometry. Data shown in all of the experiments (**A**–**D**) are representative of three independent experiments.

**Figure 3 cancers-14-02297-f003:**
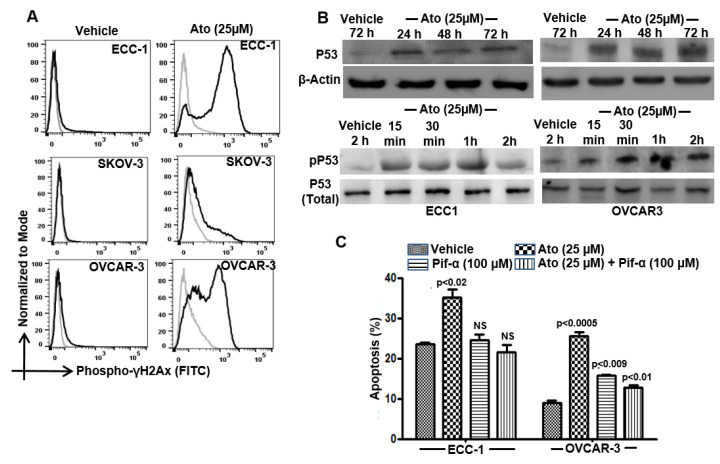
Atovaquone activates p53. (**A**) Double-strand DNA breaks in the vehicle controls and atovaquone (Ato)-treated cells were determined by intracellular flow cytometry with FITC-conjugated anti-phospho-γ-H2Ax. (**B**) Phosphorylated (Serine-15 residue) and total p53 in control and atovaquone-treated cells were monitored by Western blotting. (**C**) ECC-1 and OVCAR-3 cells were treated with atovaquone in the presence/absence of the p53 inhibitor, pifithrin-α and monitored for proliferation (MTT assay) and viability and cell death (flow cytometry for Annexin V-FITC and propidium iodide) were monitored by flow cytometry. All data shown are representative of results from three independent repeats and statistical analysis compares atovaquone-treated cells with matching controls.

**Figure 4 cancers-14-02297-f004:**
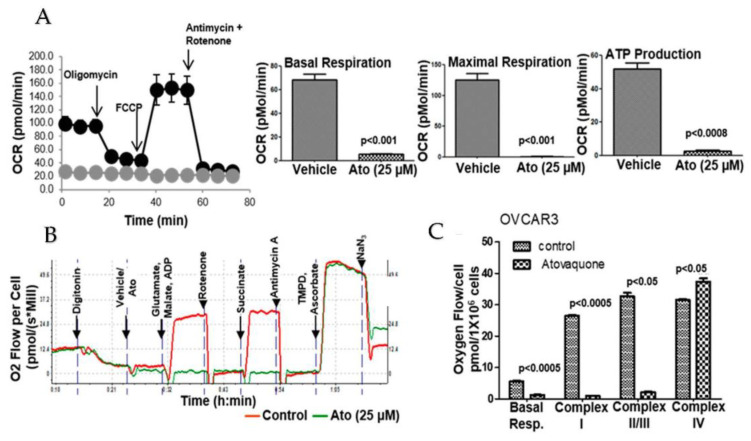
(**A**). Inhibition of oxidative phosphorylation by Atovaquone (Ato) in OVCAR-3 cells was monitored on the Seahorse Xfe96 analyzer in triplicate. Data were normalized to the number of cells plated and cell viability was confirmed at the end of the experiment. Data for ECC-1 and SKOV-3 cells are shown in Appendix A. (**B**). Complex-specific inhibition by atovaquone was monitored on an Oroboros analyzer. Digitonin was used to permeabilize the cells. Glutamate, malate and ADP enhance Complex I activity and succinate is a substrate for Complex II. TMPD and ascorbate are substrates for Complex IV. (**C**). The aggregate data from three independent experiments are shown to demonstrate that the addition of substrates for Complex I and II does not overcome the atovaquone-mediated inhibition of oxygen consumption rate (OCR).

**Figure 5 cancers-14-02297-f005:**
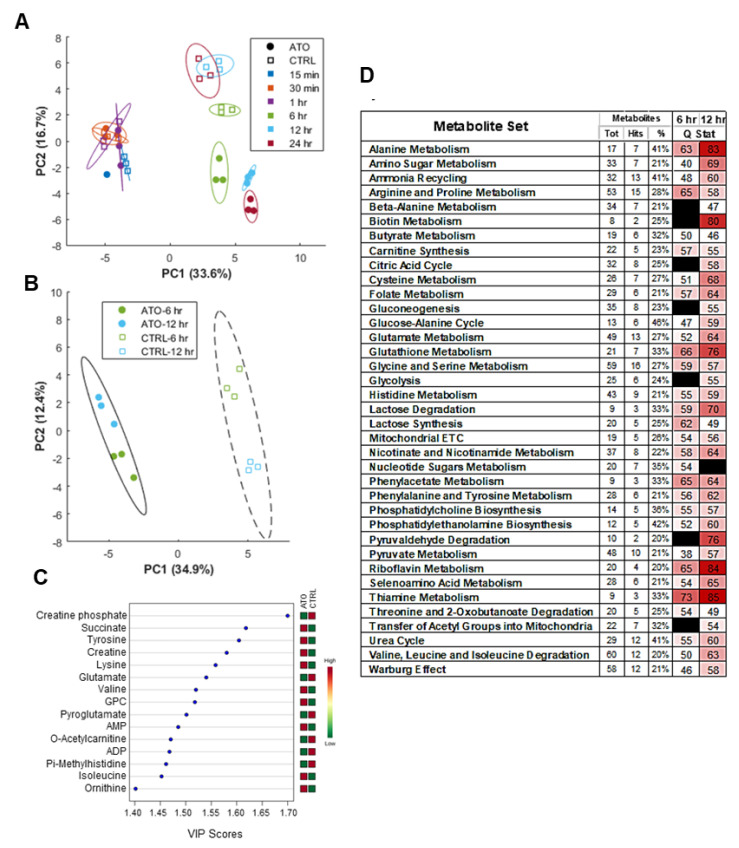
Global metabolic perturbations in response to atovaquone. ECC-1 cells were treated with atovaquone (ATO) or vehicle control (CTRL) for 0.25, 0.5, 1, 6, 12 and 24 h. 1H NMR of cell lysates and the resulting metabolite concentration data (ChenomX) was auto scaled and analyzed with MetaboAnalyst. (**A**) PCA scores plot demonstrates time and group effects, with significant treatment effect not observed until 6+ hours of exposure (each data point represents a unique replicate; bounding ellipses represent 95% confidence intervals). (**B**) PLS-DA scores plot of combined 6 and 12 h time points for ATO and CTRL groups. (**C**) Variable importance in projection (VIP) scores for the top 15 metabolites along principle component 1 of PLS-DA in panel B, which account for difference between ATO and CTRL groups. (**D**) Metabolite set enrichment analysis between ATO and CTRL groups at 6 and 12 h identifying significantly enriched pathways (Tot = total number of metabolites within annotated pathway; Hits = number of pathway metabolites present in this data set, % = percent coverage of pathway; table cutoff values of 2 hits, 20% coverage, Q-statistic > 50; black = FDR > 0.05). GPC = syn-glycero-3-phosphocholine.

**Figure 6 cancers-14-02297-f006:**
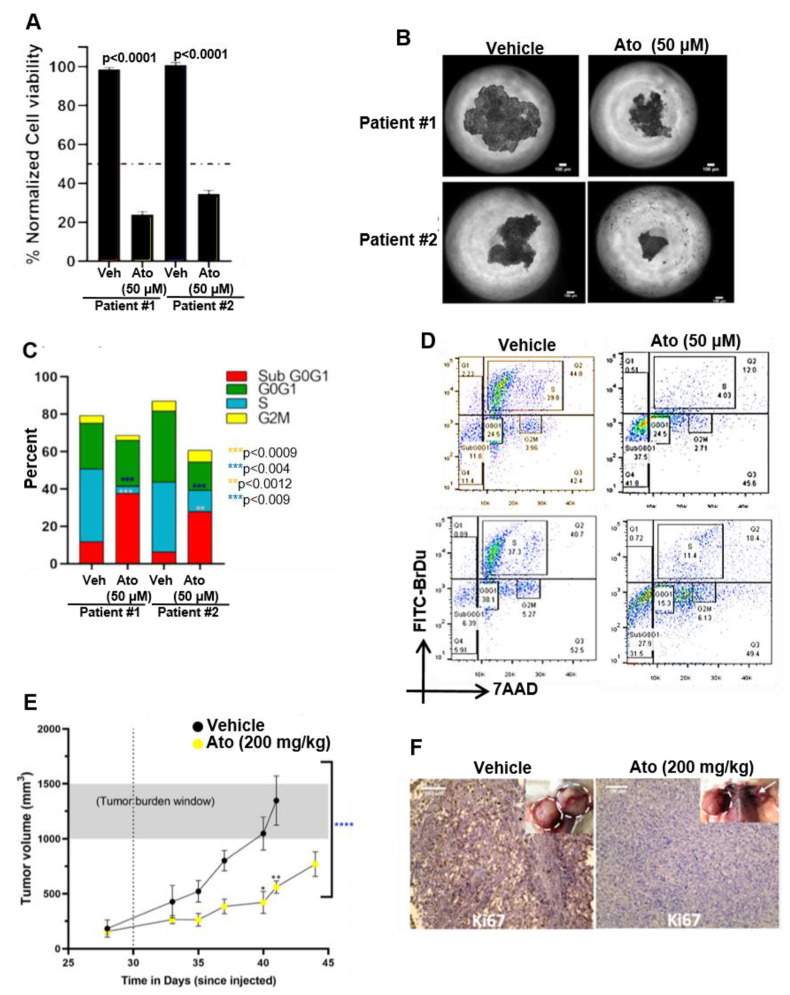
Atovaquone is effective against patient-derived ovarian cancer stem-like cells. (**A**) High-grade serous ovarian cancer patient-derived cancer stem-like cell spheroids subjected to 72 h of 50 µM atovaquone (Ato) treatment demonstrated an average of 23.5% and 35.7% viability in patients 1 and 2, respectively, with reduction in viability as compared to respective vehicle (Veh) controls (measured by MTT fluorescence). (**B**) Phase contrast demonstrates the decreased spheroid area after 72 h of 50 µM atovaquone treatment. Scale bar = 100 µm. (**C**) Cell cycle flow cytometry analysis (BrdU-FITC and 7-AAD) of spheroids subjected to 50 µM atovaquone treatment demonstrate increased population of cells undergoing apoptosis in both patient samples (Sub G0–G1 phase) as compared to the respective vehicle (Veh) controls. Additionally, a significant decrease in the population of cells in the S phase was observed in both patient samples. Collectively, these data indicate impaired DNA synthesis in response to atovaquone treatment in patient-derived ovarian cancer spheroids. (**D**) Representative cell cycle analysis for patient-derived ovarian cancer spheroids further illustrates the significant increase in Sub G0–G1 and decrease in S phase of atovaquone-treated cells. (**E**) Tumor initiation and growth kinetics are shown for NSG mice; after the tumor volume reached 100 mm3, the dotted vertical line represents the time when treatment started (day 30 after spheroids were injected). Tumor volumes are significantly different across the two groups, as analyzed by t test (n = 8, **** *p* < 0.0001, paired t test on saline control vs. atovaquone) and demonstrate major statistical significance at days 40 and 41, measured by two-way ANOVA (n = 8 mice per group, * *p* < 0.005, ** *p* < 0.01). The control group shows increased tumor burden (grey shaded area) and reached a humane end point earlier, as compared to atovaquone treatment group. (**F**) Ki67 immunohistology staining in tumor tissue sections indicates higher Ki67 expression in the saline treated tumors compared to the atovaquone treatment, matching the higher growth rate of tumors (inset) in that group. White dotted circles around the tumors in the insets display differences in tumor sizes at humane endpoint. An arrow in the inset of the atovaquone-treated group indicates a dramatic decrease in tumor size. Scale bar= 100 μm.

## Data Availability

Data are contained within the article or Appendix A.

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
