# Peer review of "Atovaquone: An Inhibitor of Oxidative Phosphorylation as Studied in Gynecologic Cancers"

_cancers, 2022, doi:10.3390/cancers14092297_

Round 1

Reviewer 1 Report

Dear Authors,

please correct the issues before publication:

Line 75- no information provided for cell culture conditions

Line 231- provide the number of approval

Lack of information about the hostile of animals, number of animals etc.

Line 233- what was the size of tumor when the treatment started, how long did it take to obtain a tumor in mice? How was it verified?

In many places the sentences lacks dots, in other it is not necessary (e.g. titles of headings, please correct it)

Line 75- no information provided for cell culture conditions

Line 231- provide the number of approval

Lack of information about the hostile of animals, number of animals etc.

Line 233- what was the size of tumor when the treatment started, how long did it take to obtain a tumor in mice? How was it verified?

In many places the sentences lacks dots, in other it is not necessary (e.g. titles of headings, please correct it)

Author Response

Thank you for the thoughtful comments.

Line 75- cell culture conditions have been added

Line 231- protocol # has been added as has information about our small animal facility, cages, and conditions

Line 233- Because these are intraperitoneal tumors, we cannot measure the size at time of injection. The tumors disperse throughout the abdominal cavity after injection (in contrast to subcutaneous tumors). Additional information about length of treatment has been included. Excised tumors (shown in Fig1E) were weighed. Experiments were stopped when the control mice developed significant ascites (palpable on exam).  

We have proofread the entire manuscript for appropriate grammar and punctuation. 

Reviewer 2 Report

Atovaquone is a drug used to treat pneumocystis. It is an inhibitor of oxidative phosphorylation and an oxygen radical generator. In this paper the authors show the expected activities of such an inhibitor in a range of experimental ovarian cancers models; in established human ovarian and endometrial cancer cell lines in culture (average IC50 10 µM); and in an orthotopic ovarian cancer model (just statistically significant) and in tumor spheroid cultures generated from ovarian cancer patient cells (given daily at 200 mg/kg orally for 15 days). The work supports the molecular-level mechanism of Atovaquone by docking studies with Mitochondrial Complexes I-III.

Overall this is a detailed, multidisciplinary, well-written study confirming Atovaquone’s mechanisms of action as a potential drug for cancer, although the numbers generated suggest it is not particularly potent.

Minor typo (line 265) commiserate with (commensurate with)

Author Response

We have corrected the minor typo in line 165 (commiserate to commensurate). 

This manuscript is a resubmission of an earlier submission. The following is a list of the peer review reports and author responses from that submission.

Round 1

Reviewer 1 Report

Dear Authors,

The manuscript presents the nice results, however its preparation is very poor: no introduction provided, lack of number of ethical approval number statements, lack of replicates, no molecular mechanistic for apoptosis provided. You should carefully revise the manuscript.

  1. Supplementary figures should be prepared as normal figure with al figure captions and explained values used with short explanation of methodology used. The figure should provide a clear results and explain it for the reader. It should be corrected before publication. The resolution of supplementary figure 3 is too low and the figure itself is not readable for the reader.
  2. Supplementary figure 5- the reference for western blot should be done on the same gels (b-actin), the is not provided the replicates of WB
  3. The introduction does not introduce the reader to the topic of the manuscript. It should explain the problem and the basic knowledge necessary to understand the results resented.
  4. Why did you use two ovary cancer cell lines and one endometrial if you tested ovarian cancer animal model?
  5. Cell lines: provide a characteristics for different cell lines used
  6. Explain how the concentrations used in cell culture might mimic the potential used in therapy
  7. Line 224: ethical aprooval numbers for animal and human study should be provided in adequeate sections
  8. Figure 1 E no statistics provided
  9. No ethical approval numbers provided

Reviewer 2 Report

This paper explores the effect of the FDA-approved anti-malarial drug atovaquone in ovarian cancer cells, both in in vitro culture and as tumor spheroids in mice. They show that it is effective in these models, and that it inhibits oxygen consumption, glycolysis, electronic transport and the citric acid cycle. They note the studies provide the mechanistic understanding and preclinical data to support further investigation of atovaquone as a cancer therapeutic. They note that they “demonstrate the anti-cancer activity of atovaquone and confirm its mechanism of action through comprehensive molecular and cell biological experiments”.

The work is well-conducted and well-reported, but it is not novel. There are more than 110 published papers on atovaquone’s anticancer properties, including two by the present authors, and three in the journal Cancers (2021, 2020 and 2018). These collectively cover all of these aspects (oxygen consumption, glycolysis, suitability for use in cancers, including ovarian).

I therefore cannot support publication unless the authors can more clearly demonstrate novelty. The future studies they note “Future studies will focus on developing a cancer-specific formulation of atovaquone and the combination of this drug with inhibitors of anti-oxidant mechanisms” would be novel.

Reviewer 3 Report

The authors' data support that Atovaquone should be paid attention to as a cancer treatment agent. The experimental design of this manuscript is reasonable and the language expression is fluent. Some minor concerns were suggested to the authors for revision.
1. It is recommended that the authors pay reasonable attention to oxidative phosphorylation as a novel cancer treatment target. For example, these two key documents are not presented in the current manuscript.
Nayak A P, Kapur A, Barroilhet L, et al. Oxidative phosphorylation: a target for novel therapeutic strategies against ovarian cancer[J]. Cancers, 2018, 10(9): 337.;
Ashton T M, McKenna W G, Kunz-Schughart L A, et al. Oxidative phosphorylation as an emerging target in cancer therapy[J]. Clinical Cancer Research, 2018, 24(11): 2482-2490.
2. The source and purity of Atovaquone need to be described in "Reagents". The reviewer thinks this is quite important piece of information.
3. It is recommended to consider providing Schrödinger authorization certificate regarding PyMOL (The PyMOL Molecular Graphics System, Version 1.7.6.0 Schrödinger, LLC.).
4. H2DCFDA-labeled ROS has some false-positive results (B. Kalyanaraman, V. Darley-Usmar, KJA Davies, PA Dennery, HJ Forman, MB Grisham, et al., Measuring reactive oxygen and nitrogen species with fluorescent probes: challenges and limitations, Free Radic. Biol. Med. 52 (2012 ) 1-6.). It is thus recommended that the authors use other probes, such as DHE, to reconfirm the key data of ROS production.
5. The current presentation of Figure 4A cannot clearly observe the interaction between the receptor and the ligand. It is recommended that the authors enlarge these figures to provide important information.
6. Figure 5D has the same concerns as above.